# International consultation on incontinence questionnaire – Urinary incontinence short form ICIQ-UI SF: Validation of its use in a Danish speaking population of municipal employees

Lærke Cecilie Grøn Jensen[ID]**[1]** *, Sidsel Boie**[2]**☯, Susanne Axelsen[ID]**[3]**☯

**1** Faculty of Health, Aarhus University, Aarhus, Denmark, **2** Department of Obstetrics and Gynaecology, Regional Hospital of Randers, Randers, Denmark, **3** Department of Obstetrics and Gynaecology, Aarhus University Hospital, Aarhus, Denmark

☯ These authors contributed equally to this work.
* laerke-cecilie@hotmail.com

**Data Availability Statement:** Data cannot be shared publicly because it contains sensitive personal information about Danish citizens. Owing

## Abstract

### Introduction

Worldwide, the estimated prevalence of urinary incontinence is 8.7%. Urinary incontinence is more frequent in women than in men. Posing the right questions is crucial, when diagnosing urinary incontinence, but also to evaluate the need of treatment and treatment effect. Therefore, reliable and validated questionnaires within this area are needed. Even though the International Consultation on Incontinence Questionnaire–Urinary Incontinence Short Form (ICIQ-UI SF) has been used on a daily basis in the Danish Urogynaecological Database since 2006, it has not yet been validated in a Danish population of both men and women.

### Objective

To test the reliability and validity of the Danish version of the ICIQ-UI SF in a Danish speaking population of men and women among municipal employees.

### Methods

Content validity was evaluated with semi-structured interviews. A quantitative field test was performed, in which the questionnaire was distributed electronically to municipal workers by E-mail. Statistical methods included item characteristics (missings, kurtosis and skewness), internal consistency (Chronbach's alfa), test-retest (ICC), construct validity (known group validation), and floor and ceiling effect.

### Results

A number of 1814 Danish municipal workers completed the questionnaire. Of the total number of responders, 426 were invited to complete the questionnaire twice (for test-retest) and

to Danish legislation, data will be available only after approval by The Danish Data Protection Agency and with a signed access agreement. Therefore, access to the data can be received following a request to forskningsprojekter@rm.dk (The Danish Data Protection Agency – Central Region of Denmark). Project number: 656336.

**Funding:** This project was financed by 'Satspuljemidler' assigned by the Danish Health Ministry to Kontinensforeningen (a Danish patient organisation). URL: https://kontinens.org The funders did not play any role in the study design, data collection and analysis decision to publish or preparation of the manuscript.

**Competing interests:** The authors have declared that no competing interests exist.

**Abbreviations:** UI, Urinary Incontinence; DUGS, Danish Urogynaecology Society; BMI, Body Mass Index; ICIQ-UI SF, International Consultation on Incontinence Questionnaire–Urinary Incontinence Short Form; REDCap, Research Electronic Data Capture; COSMIN, COnsensus-based Standards for the selection of health Measurement Instruments; ICC, Intraclass correlation coefficient; PROM, Patient reported outcome measures; GDPR, General Data Protection Regulation.

215 (50.5%) of these completed the questions again two weeks later. Statistical analyses of the ICIQ-UI SF demonstrated no floor and ceiling effects, skewness was zero and kurtosis 0.00–0.49. Cronbach's alfa was 0.87 and intraclass correlation coefficient 0.73. Two out of three hypotheses were accepted in the known-groups validation.

## Conclusion

This study offers an adaptation of the ICIQ-UI SF to a Danish setting. The Danish ICIQ-UI SF demonstrated acceptable reliability and validity. However, clinicians should consider the relatively high measurement error.

## Introduction

Worldwide, the estimated prevalence of urinary incontinence (UI) is 8.7% [1]. A meta-analysis found mean prevalence rates for men and women to be 14.5% and 23.5% respectively, as UI is more frequent in women than in men [2]. Thereby, UI is a common problem and moreover has a profoundly negative impact on health-related quality of life both physically and socially [3, 4].

When diagnosing UI, posing the right questions is crucial, and reliable and validated questionnaires are needed in clinical as well as research settings.

### The Danish ICIQ-UI SF

The International Consultation on Incontinence (ICI) recommends to use questions from their Modular Questionnaire (the ICIQ), when conducting studies on UI, to achieve international unified standardization [5]. The International Consultation on Incontinence Questionnaire–Urinary Incontinence Short Form (ICIQ-UI SF) evaluates the severity of UI symptoms and their impact on health-related quality of life. When using the ICIQ-UI SF, a total ICIQ score with a range from 0–21 is achieved from the first three questions. A score of zero means no leakage of urine and no affection on quality of life [6]. Question 1 (Q1) quantifies the frequency of urinary leaking, question 2 (Q2) evaluates the amount of leaking and question 3 (Q3) how much the urinary incontinence interferes with the everyday life.

In 1999, the questionnaire was developed by ICI sponsored by the World Health Organization in order to detect UI symptoms, their impact on quality of life and treatment outcome [5].

In Denmark, two Danish translations of the ICIQ-UI SF are used. One is approved by the ICIQ and the other is developed by the Danish Urogynaecology Society (DUGS). In accordance with previously used terminology, the version developed by DUGS will be mentioned as "VersionDUGS" [7]. VersionDUGS uses the term "leakage of urine" while the original ICIQ-UI SF uses "urinary incontinence". VersionDUGS is widely used in Denmark and has been used on a daily basis in the Danish Urogynaecological Database (containing data from women operated for UI and prolapse) since 2006 [8]. But it has not yet been validated in the general Danish population with both men and women, making it impossible for researchers and clinicians to cater the psychometric and measurement properties of the questionnaire. One validation study has been conducted on the Danish ICIQ-UI SF but this did only include women [7].

Even though validation studies of the English ICIQ-UI SF have been performed, values from Cronbach's alfa and kappa statistics range significantly and information about ceiling effect and differential item functioning are limited [9].

Since the ICIQ-UI SF is the recommended first choice instrument, validity and reliability are still continuously relevant to evaluate, and especially when using the instrument on a new study population never tested with the ICIQ-UI SF before. Evaluation of validity and reliability of the Danish ICIQ-UI SF has never been performed in a large study sample of different sexes or in the general Danish population, even though UI affects all sexes and age groups.

The aim of this study was to test the validity and reliability of the Danish translated version of the ICIQ-UI SF, VersionDUGS, in a population reflecting the general Danish population.

UI is not only a problem among women and the Danish translated version of ICIQ-UI SF, VersionDUGS, is not only used within gynaecology but widely used within different specialities such as general practitioners, urology, neurology, and physiotherapy. Therefore, is it highly relevant with an evaluation of the validity and reliability in a big study sample with different sexes, a wide age distribution and a diverse education level. This study provides clinicians and researchers with such an evaluation.

## Methods

### The questionnaire

Additional questions were included to supplement the ICIQ-UI SF with background data and information about chronic diseases. Background data include information about age, sex, Body Mass Index, education level, smoking habits, highest completed education, civil status, and chronic diseases. The questionnaire is shown in S1 Table. The questionnaire was created in and distributed from REDCap—Research Electronic Data Capture, a secure web-based platform for construction and management of surveys and online databases [10].

The original ICIQ-UI SF in English and all translated versions can be found on ICIQ's official webpage and all copyrights of these are preserved by ICIQ [11].

Approval to use VersionDUGS was provided in a written form from DUGS.

### Pilot test: Content validity

VersionDUGS was pilot tested to evaluate face validity and content validity. The COSMIN panel defines face validity as 'the degree to which a measurement instrument, indeed, looks as though it is an adequate reflection of the construct to be measured', and the content validity as 'the degree to which the content of a measurement instrument is an adequate reflection of the construct to be measured' [12]. In accordance with the COSMIN checklist, content validity was evaluated with a pilot test on a small sample size from a relevant population—in this case 14 public workers (nine women and five men, from 27 to 63 years old).

Pilot-testing was performed with semi-structured interviews with the 14 public workers during and after they had answered the questionnaire. Participants to the pilot-test were recruited with informative emails to employees in a specific department with permission from the management. The 'Three-step Test-interview' (TSTI) was the basis for the semi-structured interviews. It combines the 'think loud' and 'probing' methods, which makes it a powerful method for evaluating the comprehension and comprehensibility [12]. The three steps in the TSTI consist of: 1) Observational data through concurrent think loud; observation of the respondent's behaviour while filling in the questionnaire and asking the respondent to 'think loud', 2) Focused interview–clarifying the observed; the interviewer asks with the purpose of filling in gaps of the observational data and 3) Semi-structured interview–eliciting experiences; questions about response behaviour, wording in the questionnaire and understanding of definitions [13].

The participants were asked about the relevance of each item, the comprehensiveness of the questionnaire, the comprehensibility of the instructions, items and response options

and lastly, if they missed any items or response options. Furthermore, six medical professionals within the area of focus were asked about the relevance of each item and the comprehensiveness.

## Quantitative field test of the ICIQ-UI SF

The validity and reliability of VersionDUGS were investigated in a quantitative field-test. Participants were recruited through the municipal of their workplace. A formal invitation to participate including a study presentation and instructions were sent to the directors of the municipal. The directors were asked to distribute an email invitation with the open link questionnaire together with information about data storage and GDPR to their municipal employees. Several of the participating municipals distributed the invitation internally. For this reason, it is not possible to report the response rate. Inclusion criteria were age over 18 years and municipal employment. In the bottom of the questionnaire, we asked for their permission to send them the same questionnaire once again (for test-retest).

## Statistical analysis

STATA was used for statistical analyses. Statistical methods include item characteristics (missings, kurtosis and skewness), internal consistency (Cronbach's alfa), test-retest (ICC), construct validity (known group validation), and floor and ceiling effect.

**Item characteristics and floor and ceiling effects.** Skewness and kurtosis are both statistical measures for distribution and reveal if answers are normal distributed variables. If skewness equals zero it reflects a normal distribution while a negative skewness reflects a left-skewed distribution where the mean is lower than the median [14]. If kurtosis equals three it reflects a normal distribution of the variable, while higher than three is called leptokurtic and below three platykurtic. A leptokurtic distribution (high kurtosis) is characterized by a certain amount of peakedness while a playtykurtic distribution (low kurtosis) is characterized by a certain amount of flatness with fewer and less extreme outliers than normal distributed variables [15].

**Internal consistency.** The COSMIN panel defines internal consistency as the interrelatedness between items and is only relevant to perform on patient reported outcome measures of the reflective model and when all items form a unidimensional scale [16]. Internal consistency is calculated for Q1-Q3 in the ICIQ-UI SF with Cronbach's alfa [17].

**Reliability—test-retest.** Intraclass correlation coefficient (agreement) was calculated for Q1–Q3 in the ICIQ-UI SF [18]. An ICC under 0,5 is poor, between 0.5–0.75 moderate, over 0.75 good and over 0.9 is considered excellent [19]. Standard error of measurement-agreement (SEM$_{agreement}$) and then the smallest real different (SRD) was calculated with the following equations:

$$SEM = SD \sqrt{1 - ICC}. \quad SD = (SD_{time1} + SD_{time2})/2.$$

$$SRD = SEM * 1.96 * \sqrt{2}.$$

**Construct validity.** Three hypotheses were tested: 1) women are more likely to be urinary incontinent than men [20], 2) participants older than 50 years are more likely to be urinary incontinent than participants younger than 50 years [21], 3) participants with a BMI≥30 are more likely to be urinary incontinent than participants with BMI<30 [22, 23].

### Ethics

The project was approved by the Danish Data Agency (656336) and The National Ethical Committee has approved that this project was carried out without their involvement (1-10-72-186-19).

Consent to participate in the evaluation was achieved in a written enquiry. We sent the information about the evaluation to the directors of the participating municipals and asked them if they would allow their employees to participate. If they accepted the participation, the same information about purpose, questionnaire, time consumption, GDPR rules, anonymization of the responder, data protection and -storage, ethics of the study and that participation could be interrupted at any time was sent to the employees.

## Results

### Interview findings: Content validity

Interview participants found the ICIQ-UI SF comprehensive and easy to complete. Therefore, the three step test interviews did not lead to any changes.

### Field test

The questionnaire was electronically distributed to municipal workers in 16 Danish municipals from 20.01.2020 to 11.05.2020. A total of 1825 persons opened the questionnaire, but 11 of these did not complete any items and were excluded. Therefore, the final study sample consisted of 1814 participants. Among the responders, 426 were invited to answer the questionnaire again after two weeks and 215 of these answered and were included in test-retest (response rate for second round = 50.5%). A total of 418 of the 1814 responders (23%) reported urinary incontinence. The participants reported demographics as shown in Table 1.

### ICIQ-UI SF, VersionDUGS

**Item characteristics and floor and ceiling effects.** Floor effect of the ICIQ-score was 4.26% and ceiling effect was 0.24%, indicating that the ICIQ-UI SF manages to differentiate between the respondents.

Skewness was close to zero in Q1-Q3, while kurtosis was 2.79 in Q1, 13.28 in Q2 and 2.88 in Q3 as shown in Table 2.

The skewness of Q1-Q3 is positive but close to zero and thereby indicates a distribution of the answers close to a normal distribution but slightly right-skewed. The kurtosis of Q1 and Q3 indicates a platykurtic distribution with fewer and less extreme outliers than a normal distribution [15]. The kurtosis of Q2 = 13.28 is per definition a leptokurtic distribution, which produces more outliers than a normal distribution.

**Internal consistency.** Cronbach's alfa for Q1-Q3 was 0.87.

**Reliability.** Of 426 invited respondents, 215 completed the questionnaire again after two weeks for test-retest (response rate = 50.47%). The intraclass correlation coefficient (agreement) was calculated for these participants and was 0.73 for Q1-Q3. $SEM_{agreement}$ was calculated: $SEM = 3.43 \sqrt{1 - 0.73} = 1.78$

$$SRD = 1.78 * 1.96 * \sqrt{2} = 4.9.$$

**Construct validity–known-groups validation.** Risk differences were calculated to test three hypotheses. Two out of three hypotheses were accepted: women are more likely to have incontinence than men, RD = 0.22 (95%-CI:0.17–0.25) and participants with a

**Table 1. Baseline demographics.**

|  | All | Incontinence, yes |
|---|---|---|
|  | N = 1814 | N = 418 (23%) |
| Age, years mean (SD) | 48.30 (11.43) | 50.07 (9.8) |
| Sex, female n (%) | 1402 (83.8) | 396 (88.3) |
| BMI, kg/cm$^2$ mean (SD) | 26.6 (5.4) | 27.9 (6.0) |
| Highest completed Education n (%) |  |  |
| Primary school | 35 (2.1) | 0 (0) |
| Secondary education | 32 (1.9) | 5 (1.2) |
| Skilled | 198 (11.8) | 45 (10.8) |
| Short higher education | 292 (17.4) | 76 (18.2) |
| Medium higher education | 851 (50.8) | 223 (53.3) |
| Long higher education | 243 (14.5) | 58 (13.9) |
| Other | 23 (1.4) | 11 (2.6) |
| Civil status n (%) |  |  |
| Single | 246 (14.7) | 51 (12.2) |
| Married or cohabitating | 1346 (74.2) | 352 (84.2) |
| Other | 71 (4.3) | 15 (3.6) |
| Smoking, yes n (%) | 236 (14.2) | 48 (11.5) |
| Chronic diseases[*], yes n (%) | 437 (26.1) | 129 (30.9) |

[*]Chronic diseases were exemplified in the questionnaire with diabetes, rheumatoid arthritis, chronic obstructive pulmonary disease and heart diseases.

BMI≥30 are more likely to have incontinence than participants with BMI<30, RD = 0.11 (95%-CI: 0.06–0.16).

## Discussion

This is the first validation study of the Danish translated version of ICIQ-UI SF, VersionDUGS in a population reflecting the general Danish population.

Interview-participants found the questions appropriate, comprehensive, and easy to complete.

We had 0 missing answers for Q1 and Q2, and 1.2% for Q3 which is consistent with the number of missings in other validation studies [7, 24]. Floor and ceiling effects as well as skewness and kurtosis were acceptable in our study group.

The Cronbach's alfa was high (0.87) compared to the validation study conducted by Clausen et al. reporting a Cronbach's' alfa of 0.7 [7]. Cronbach's alfa ranges from 0.71–0.78 in validation studies of the ICIQ-UI SF in languages such as Persian, Chinese, and Japanese [25–27], while higher values of Cronbach's alfa has been shown in validation studies from Croatia and Slovenia on 0.85 and 0.81, respectively [28, 29]. Cronbach's alfa is highly affected by the variation of the population and participants in a heterogenous population will have a higher

**Table 2. Skewness and Kurtosis for Q1-Q3.**

|  | Observations | Mean (SD) | Skewness | Kurtosis | Missing, n (%) |
|---|---|---|---|---|---|
| Q1 | 418 | 1.7 (1.1) | 0.89 | 2.79 | 0 |
| Q2 | 418 | 2.0 (0.6) | 0.83 | 13.28 | 0 |
| Q3 | 413 | 3.2 (2.5) | 0.89 | 2.88 | 5 (1.2) |

Cronbach's alfa than participants in a more homogenous population [30]. Our population consists of different sexes, have a wide age distribution, and represents a broad level of education, making it a very heterogenous population. This could explain why VersionDUGS has a higher Cronbach's alfa in our validation study than in the Danish validation study by Clausen et al. [7].

The intraclass correlation coefficient (ICC) of 0.73 is moderate. An ICC of 0.73 is acceptable and indicates that responders of the Danish translated version of ICIQ-UI SF, VersionDUGS did not differ their answers in a two weeks' time period [31]. Clausen et al. also showed stable test-retest results in a group consisting of Danish women and studies from other countries show test-retest results from acceptable to excellent [5, 7, 32]. While ICC is unitless, the SEM of 1.78 has the same unit as the measurement score. The SRD of 4.9 was calculated from the SEM and tells us that an individual's difference on repeated testing on 4.9 or greater will reflect a real difference in 95% of the cases. This is a relatively high SRD, indicating high measurement error which clinicians may want to consider, when using VersionDUGS.

Finally, two out of three hypotheses were accepted in the known-groups validation.

## Strengths and limitations

As in any qualitative study, interview findings in the pilot test of the questionnaire could be biased by the interviewer's preunderstanding of the field. However, using a specific interview model (TSTI) and making the pilot test structured and systematic helped reducing this type of bias [12, 13]. Being conscious of our preunderstanding when evaluating the interviews decreased any potential bias as well. Finally, adopting the relevant design criteria of the COSMIN checklist in the quantitative field test of the ICIQ-UI SF reduced any idiosyncrasies [33, 34]. Nevertheless, the quantitative field test had some possible limitations. Only municipal workers were included, which should be taken into account when using VersionDUGS in other populations. This may affect the generalization of our results to an arbitrary population. However, the large study group had no other specific characteristics making them less representative. Moreover, it is by far the largest study group of different sexes in a validation study of the ICIQ-UI SF. Unfortunately, due to the open link invitation, it was not possible to estimate response rates or evaluate if the non-responders differed in terms of baseline characteristics from the responders. Nevertheless, the recruitment method is justified by the goal of achieving a large study group with a broad age distribution and the fact that no other opportunities were available when targeting municipal workers.

## Conclusion

The Danish translated version of ICIQ UI SF, VersionDUGS is a valid and reliable measure of urinary incontinence in a Danish population consisting of different sexes. However, clinicians should consider the relatively high measurement error of VersionDUGS.

## Supporting information

**S1 Table. The questionnaire.**
(DOCX)

## Author Contributions

**Data curation:** Lærke Cecilie Grøn Jensen.

**Formal analysis:** Lærke Cecilie Grøn Jensen.

**Funding acquisition:** Susanne Axelsen.

**Investigation:** Lærke Cecilie Grøn Jensen.

**Methodology:** Sidsel Boie.

**Project administration:** Lærke Cecilie Grøn Jensen, Susanne Axelsen.

**Supervision:** Sidsel Boie, Susanne Axelsen.

**Validation:** Sidsel Boie.

**Writing – original draft:** Lærke Cecilie Grøn Jensen.

**Writing – review & editing:** Sidsel Boie, Susanne Axelsen.

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
