## [Decision Letter · Decision Letter 0]

26 Nov 2021

PONE-D-21-30384International Consultation on Incontinence Questionnaire – Urinary Incontinence Short Form ICIQ-UI SF: Validation of its use in a Danish speaking population of municipal employeesPLOS ONE

Dear Dr. Grøn Jensen,

Thank you for submitting your manuscript to PLOS ONE. After careful consideration, we feel that it has merit but does not fully meet PLOS ONE’s publication criteria as it currently stands. Therefore, we invite you to submit a revised version of the manuscript that addresses the points raised during the review process.

ACADEMIC EDITOR: Apart from the reviewers comments I ask you to consider that the readers may want to know how your analysis of the questionnaire relates to the original, eg are floor and ceiling effects, consistency and or test retest variation (very) different from the original, or not?

We look forward to receiving your revised manuscript.

Kind regards,

Peter F.W.M. Rosier, M.D. PhD

Academic Editor

PLOS ONE

Journal Requirements:

Reviewers' comments:

Reviewer's Responses to Questions

**Comments to the Author**

1. Is the manuscript technically sound, and do the data support the conclusions?

Reviewer #1: Yes

Reviewer #2: Yes

2. Has the statistical analysis been performed appropriately and rigorously? 

Reviewer #1: Yes

Reviewer #2: Yes

3. Have the authors made all data underlying the findings in their manuscript fully available?

Reviewer #1: Yes

Reviewer #2: Yes

4. Is the manuscript presented in an intelligible fashion and written in standard English?

Reviewer #1: Yes

Reviewer #2: No

5. Review Comments to the Author

Reviewer #1: Thank you for the opportunity to peer review your manuscript

In this study you have performed a validation of a Danish version of the International Consultation on Incontinence Questionnaire-urinary incontinence-short form (ICIQ-UI-SF) named VersionDUGS using Classical Test Theory (TCC).

In a pilot test with 14 public workers and six health care professionals you have performed thorough qualitative analyses of both face and content validity and construct validity has been evaluated through three hypotheses (known-group validity) (sex, age, BMI). Furthermore, you have examined internal consistency of the three items that can be scored with the Cronbach’s alfa.

Pretesting (field test) of the questionnaire has been performed in male and female municipal workers who have been invited through email to complete an online version of the questionnaire

Test-retest reliability has been examined by calculating an Intraclass correlation coefficient (ICC)

You find good face and content validity and two of your three hypotheses are confirmed (Sex and BMI).

1814 municipal workers (1402 female), have completed the questionnaire and 215 (50.5%) of 426 invited complete it twice

You find low internal consistency of the three items (0.48) while floor and ceiling effect is low. Distribution of scores is adequate.

Test-retest reliability based on the ICC is moderate (0.73)

You conclude that the Danish VersionDUGS has acceptable validity and reliability

Major comment

This is a well performed study with an adequate sample size. You have followed the COSMIN checklist and fulfill the criteria for a high-quality study. Your methods are well described including your qualitive method which is very well explained.

You describe that there are two versions of the ICIQ-UI SF in Danish, and that the VersionDUGS is widely used in a Danish database but not approved by the ICIQ (Bristol Institute). Could it be a problem to have two versions of the ICIQ-UI SF in Danish? Did you consider comparing the two versions in a study? Would it be possible to merge the two versions?

You determine relative reliability using ICC and your ICC is only moderate. It would be of interest to the reader know the absolute reliability: I.e. standard error of measurement (SEM 95%), Smallest real difference (SRD) or minimal detectable change (MDC). Alternatively, you could present limits of agreement (Bland and Altman method)

Minor comments

Gaussian is an old term, for the reader the term “normal distribution” would be preferred

P. 10, l. 187: The ICC model: 2.3 agreement, where does the “3” come from?

How were the participants for the pilot test recruited?

How did you recruit respondents for the retest? Randomly?

You find a low Cronbach´s alfa but do not describe the value of each item? Probably, the low alfa is not a problem since number of items has an influence on this. Even though inter-item correlation may be higher if you add more items as you suggest in your discussion this may not improve content validity.

Reviewer #2: I congratulate the authors for composing a work towards the use of a valid version

of the ICIQ in Denmark. I'm certain this will expand their possibilities to study the

outcomes of their patients in a more reliable way. Authors often state that a quality of life scale is

''valid''. However, these instruments should not be viewed as valid or not valid since validity and

reliability are based on evidence accumulated over time, not one-time, dichotomous concepts.

Validation studies attempt to measure the extent to which evidence supports that the instruments

are measuring what they are supposed to measure and quantitate the error in doing so. Based on

this, the study is useful since it demonstrated that the instruments performed as designed in a

different population with a different language.

I suggest improving discussion with some of the validation studies which have demonstrated higher Cronbach-alpha value - such as: Mikuš M, Ćorić M, Matak L, Škegro B, Vujić G, Banović V. Validation of the UDI-6 and the ICIQ-UI SF - Croatian version. Int Urogynecol J. 2020 Dec;31(12):2625-2630. doi: 10.1007/s00192-020-04500-4. Epub 2020 Aug 21. PMID: 32821964. AND Rotar M, Tršinar B, Kisner K, et al. Correlations between the ICIQ- UI short form and urodynamic diagnosis. Neurourol Urodyn. 2009;28(6):501–5.

Also, there is need for improvement in English style and overall language.

6. PLOS authors have the option to publish the peer review history of their article (what does this mean?). If published, this will include your full peer review and any attached files.

Reviewer #1: No

Reviewer #2: No

---

## [Author Response · Author response to Decision Letter 0]

22 Jan 2022

Dear Mr. Rosier, 

We thank you as well as the reviewers for your time, your expertise and your useful valuable comments to improve our manuscript. 

 In accordance with your inquiries, we have: 

- Checked and updated our submission in relation to PLOS ONE’s style requirements. 

- Provided additional information about consent in the Ethics section and in the online formula.

- Revised the reference list. 

In the following, we answer each comment from Reviewer 1 and 2, starting with the major comments and then the minor. However, first we have one major change in the manuscript, that we would like to account for: 

In relation to the revision, I recalculated all the statistics of the results, which led to the discovery of a incorrect item which was included in the previous calculation of Cronbach’s alfa. I recalculated the Cronbach’s alfa for the three correct items (included in the ICIQ-score) and discussed the result with my supervisors/co-authors. 

Of course, we prefer transparency about our incorrect calculation included in the first manuscript draft and have now corrected the result of Cronbach’s alfa and the relevant paragraphs surrounding the Cronbach’s alfa in the results, discussion and conclusion sections. 

We hope that our honesty and scrupulous revision will be well received and appreciated. 

Reviewer 1:

#Major comment 1: Could it be a problem to have two versions of the ICIQ-UI SF in Danish? Did you consider comparing the two versions in a study? Would it be possible to merge the two versions?

VersionDUGS is widely used in Denmark and especially within the field of gynecology, where it is used for reporting to the Danish database (DugaBase). 

The difference between the two versions does not involve the three items used for the ICIQ-score. Therefore, and for practical reasons, we decided only to validate VersionDUGS. 

VersionDUGS is validated for women by Clausen et al. and we continued by validating VersionDUGS in a population with both men and women. 

#Major comment 2: It would be of interest to the reader know the absolute reliability. 

We have added Standard Error of Measurement to supplement our ICC. 

Answer to minor comments: 

- “Gaussian distribution” is changed to “Normal distribution” 

- “2.3 agreement” has been replaced with just “ICC (agreement)” since the “2.3” information won’t be useful to the readers of the article. 

- Information is added about recruitment to both the pilot test and the retest. 

Reviewer 2: 

#Major comment 1: I suggest improving discussion with some of the validation studies which have demonstrated higher Cronbach-alpha value. 

We have included validation studies with higher values of Cronbach’s alfa as requested (page 11 line 217). 

#Major comment 2: Also, there is need for improvement in English style and overall language.

We have to our very best tried to improve the overall language and the manuscript was proofread by a medically knowledgeable with English as his mother tongue. 

All changes are marked with yellow in the manuscript with track changes. 

We look forward to hearing from you regarding our submission. We will thorughly respond to any further questions or comments that you might have. 

Kind regards on behalf of the author group 

Lærke Cecilie Grøn Jensen

Medical Student, Faculty of Health, Aarhus University, Denmark

---

## [Decision Letter · Decision Letter 1]

10 Feb 2022

PONE-D-21-30384R1International Consultation on Incontinence Questionnaire – Urinary Incontinence Short Form ICIQ-UI SF: Validation of its use in a Danish speaking population of municipal employeesPLOS ONE

Dear Dr. Grøn Jensen,

Thank you for submitting your manuscript to PLOS ONE. After careful consideration, we feel that it has merit but does not fully meet PLOS ONE’s publication criteria as it currently stands. Therefore, we invite you to submit a revised version of the manuscript that addresses the points raised during the review process.

ACADEMIC EDITOR: A small challenge remains, can you add the information to the manuscript?==============================

We look forward to receiving your revised manuscript.

Kind regards,

Peter F.W.M. Rosier, M.D. PhD

Academic Editor

PLOS ONE

Journal Requirements:

Additional Editor Comments (if provided):

None

Reviewers' comments:

Reviewer's Responses to Questions

**Comments to the Author**

1. If the authors have adequately addressed your comments raised in a previous round of review and you feel that this manuscript is now acceptable for publication, you may indicate that here to bypass the “Comments to the Author” section, enter your conflict of interest statement in the “Confidential to Editor” section, and submit your "Accept" recommendation.

Reviewer #1: (No Response)

Reviewer #2: All comments have been addressed

2. Is the manuscript technically sound, and do the data support the conclusions?

Reviewer #1: Partly

Reviewer #2: Yes

3. Has the statistical analysis been performed appropriately and rigorously? 

Reviewer #1: No

Reviewer #2: Yes

4. Have the authors made all data underlying the findings in their manuscript fully available?

Reviewer #1: Yes

Reviewer #2: Yes

5. Is the manuscript presented in an intelligible fashion and written in standard English?

Reviewer #1: Yes

Reviewer #2: Yes

6. Review Comments to the Author

Reviewer #1: Thank you for your revised manuscript

As requested, you have provided the equation for the calculation of absolute reliability (Standard Error of Measurement (SEM)) and you also very shortly report the result from your calculation, but you have not explained what it means in relation to the reliability of the ICIQ-VersionDUGS.

It seems that you have calculated a SEM (68% at group level), but you do not tell what 1.57 means to the test-retest reliability of the ICIQ-VersionDUGS. Furthermore, this figure would be 3.1, if you had calculated SEM (95% at group level), which is relevant for research settings (SEM x1.96), and 4.3 at an individual level. The latter representing the smallest real difference (SRD) (SEM x 1.96 x √2).

The implications of the results above should be described and discussed since it is important knowledge for both researchers and clinicians.

In your discussion page 12, line 234 you write:” The intraclass correlation coefficient of 0.73 is moderate, which is confirmed in the calculated standard error of measurement”. Why does the ICC affect SEM and could you have calculated the SEM based on another analysis (ANOVA)?

Line 235: should be a zero.

Reviewer #2: Thank you for the opportunity to re-revise this manuscript. The revised version is fully acceptable for publication in PLOS ONE.

7. PLOS authors have the option to publish the peer review history of their article (what does this mean?). If published, this will include your full peer review and any attached files.

Reviewer #1: No

Reviewer #2: No

---

## [Author Response · Author response to Decision Letter 1]

17 Mar 2022

Peter F.W.M. Rosier, M.D. PhD

Academic Editor

PLOS ONE

Resubmission date: 17.03.2022

Dear Mr. Rosier, 

We thank you as well as the reviewers for your time, your expertise and your useful valuable comments to improve our manuscript. We are very grateful to the reviewers for their comments on our manuscript. 

All changes are marked with yellow in the manuscript with track changes and we hope that these revisions are sufficient to make our manuscript suitable for publication in PLOS ONE. 

Here is a point-by-point response to the reviewers’ comments: 

Reviewer 1:

Thank you for your revised manuscript

As requested, you have provided the equation for the calculation of absolute reliability (Standard Error of Measurement (SEM)) and you also very shortly report the result from your calculation, but you have not explained what it means in relation to the reliability of the ICIQ-VersionDUGS.

It seems that you have calculated a SEM (68% at group level), but you do not tell what 1.57 means to the test-retest reliability of the ICIQ-VersionDUGS. Furthermore, this figure would be 3.1, if you had calculated SEM (95% at group level), which is relevant for research settings (SEM x1.96), and 4.3 at an individual level. The latter representing the smallest real difference (SRD) (SEM x 1.96 x √2).

The implications of the results above should be described and discussed since it is important knowledge for both researchers and clinicians.

In your discussion page 12, line 234 you write:” The intraclass correlation coefficient of 0.73 is moderate, which is confirmed in the calculated standard error of measurement”. Why does the ICC affect SEM and could you have calculated the SEM based on another analysis (ANOVA)?

Line 235: should be a zero.

Answer to reviewer 1:

We thank reviewer 1 for the valuable suggestions and we agree that we could improve our use and presentation of SEM. We have therefore been in contact with a Danish expert in questionnaire technique and clinemetrics to ensure that we use the appropriate equations in our validation study. 

We were guided to use the following equations to calculate the appropriate SEM to our ICC: 

SEM=SD √(1-ICC) SD=(SD_time1+SD_time2)/2. 

We include these equations and the calculations in the manuscript to make the results transparent and to enable readers of our manuscript to calculate further on the numbers. 

As you suggested, we furthermore calculated the smallest real difference. A comment on the SEM and SRD is added to the discussion (p. 13 line 233) as well as in the conclusion (p. 14 line 257). 

Reviewer 2: Thank you for the opportunity to re-revise this manuscript. The revised version is fully acceptable for publication in PLOS ONE

Answer to reviewer 2: 

Thank you very much for this comment. We are happy to read that you find our manuscript acceptable for publication. 

We look forward to hearing from you and will happily respond to any further questions or comments you may have. 

Kind regards on behalf of the author group 

Lærke Cecilie Grøn Jensen

Medical Student, Faculty of Health, Aarhus University, Denmark 

Vennelyst Boulevard 4, 8000 Aarhus C 

laerke-cecilie@hotmail.com

Tel.: +45 30 69 96 74

---

## [Editor Report · Decision Letter 2]

22 Mar 2022

International Consultation on Incontinence Questionnaire – Urinary Incontinence Short Form ICIQ-UI SF: Validation of its use in a Danish speaking population of municipal employees

PONE-D-21-30384R2

Dear Dr. Grøn Jensen,

We’re pleased to inform you that your manuscript has been judged scientifically suitable for publication and will be formally accepted for publication once it meets all outstanding technical requirements.

Kind regards,

Peter F.W.M. Rosier, M.D. PhD

Academic Editor

PLOS ONE
---

## [Editor Report · Acceptance letter]

28 Mar 2022

PONE-D-21-30384R2 

International Consultation on Incontinence Questionnaire – Urinary Incontinence Short Form ICIQ-UI SF: Validation of its use in a Danish speaking population of municipal employees 

Dear Dr. Grøn Jensen:

I'm pleased to inform you that your manuscript has been deemed suitable for publication in PLOS ONE. Congratulations! Your manuscript is now with our production department. 

Kind regards, 

on behalf of

Dr. Peter F.W.M. Rosier 

Academic Editor

PLOS ONE